# Mussel-Inspired Lego Approach for Controlling the Wettability of Surfaces with Colorless Coatings

**DOI:** 10.3390/biomimetics8010003

**Published:** 2022-12-21

**Authors:** Carolina Casagualda, Juan Mancebo-Aracil, Miguel Moreno-Villaécija, Alba López-Moral, Ramon Alibés, Félix Busqué, Daniel Ruiz-Molina

**Affiliations:** 1Catalan Institute of Nanoscience and Nanotechnology (ICN2), CSIC and The Barcelona Institute of Science and Technology (BIST), Campus UAB, Bellaterra, 08193 Barcelona, Spain; 2Departament de Química, Universitat Autònoma de Barcelona, Bellaterra, 08193 Barcelona, Spain; 3Instituto de Química del Sur-INQUISUR (UNS-CONICET), Universidad Nacional del Sur, Bahía Blanca 8000, Argentina

**Keywords:** bioinspired, catechol, coatings, multifunctional, oil spill

## Abstract

The control of surface wettability with polyphenol coatings has been at the forefront of materials research since the late 1990s, when robust underwater adhesion was linked to the presence of L-DOPA—a catecholic amino acid—in unusually high amounts, in the sequences of several mussel foot proteins. Since then, several successful approaches have been reported, although a common undesired feature of most of them is the presence of a remnant color and/or the intrinsic difficulty in fine-tuning and controlling the hydrophobic character. We report here a new family of functional catechol-based coatings, grounded in the oxidative condensation of readily available pyrocatechol and thiol-capped functional moieties. The presence of at least two additional thiol groups in their structure allows for polymerization through the formation of disulfide bonds. The synthetic flexibility, together with its modular character, allowed us to: (I) develop coatings with applications exemplified by textiles for oil-spill water treatment; (II) develop multifunctional coatings, and (III) fine-tune the WCA for flat and textile surfaces. All of this was achieved with the application of colorless coatings.

## 1. Introduction

Chemically engineered interfaces regulate the interaction with the nearby environment by endorsing functional properties such as chemical inertness, adhesion, biocompatibility and/or hydrophilicity/hydrophobicity, among others [1,2,3,4,5]. Most of the coatings so far reported, either as a polymer or self-assembled monolayer, Refs. [6,7,8,9] rely on specific chemical interactions between a given substrate and the material used as a coating. More recently, the development of substrate-independent functional thin films such as organic polydopamine (PDA) [10,11,12,13,14,15] coatings and hybrid metal–phenolic networks (MPNs), [16,17,18,19,20,21] have attracted widespread interest, owing to their universal adhesion and high stability. Such characteristics are all imparted by the incorporation of polyphenol-containing building blocks inspired by the strong adhesion of several marine organisms, such as mussels and sandcastle worms. Those species adhere to virtually any substrate in aqueous environments, and this is directly associated with the secretion of proteinaceous substances that contain a high concentration of the catecholic amino acid L-DOPA [22]. This compound can anchor by different mechanisms involving covalent bonds, coordination chemistry, π-π stacking, π-cation, electrostatic or hydrogen bonds [23]. Among the different functions that these coatings have been used for, the number of catechol-based examples targeted to systematically control and modify the wettability of surfaces are numbered [24,25,26,27,28,29]. This represents nowadays one of the biggest challenges in materials science, owing to its significance in many relevant application areas such as anti-corrosion and -frozen coatings, hydrophobic and stain-resistant textiles, or oil spill-treatment, inter alia. One of the most straightforward approaches used is the self-polymerization under alkaline conditions of catechol-based compounds, such as dopamine or norepinephrine, and subsequently the functionalization with alkyl chains [30,31,32,33]. Alternatively, catechol-based molecules can be first functionalized with alkyl chains and then polymerized under alkaline media [34,35,36,37,38]. In a second approach, catechol units can be grafted onto an already formed polymer with additional functional groups that determines the wettability of the surface [39,40]. And finally, a third approach which has not been very much exploited consists of polymerization through a functionality present on the catechol (e.g., acrylate group) that allows for a higher degree of functionalization and synthetic control on the final structure [41,42]. However, despite the different examples so far reported, there are no cases where this property could be systematically fine-tuned at will over a broad range of water contact angles (WCA). Another important fact to mention is that most of the successful examples so far reported in the literature involve the coloration of the substrate, which is not desired in most cases. Therefore, the development of novel synthetic approaches that simultaneously overcome both relevant limitations are strongly required. 

Herein, we report the design and synthesis of a new family of colorless coatings with fine-tuned hydrophobic character that relies on a catechol-grafted polymeric architecture linked through disulfide bridges (Figure 1). The basic scaffold for the modular design of all monomers reported in this work is pentaerythritol tetrakis (3-mercaptopropionate) **1**, a cross-linking agent widely used in commercial coatings, adhesives and sealants. The catechol conjugation is achieved through the addition of one of the thiol groups to the corresponding oxidized *o*-quinone, inspired by the alleged role of free thiol groups of cysteine residues in mussel foot proteins [43,44,45,46]. A second thiol group allows for the simultaneous incorporation of an alkyl chain bearing a terminal vinyl or acrylate group using a thiol-ene click reaction [47,48,49]. And finally, the last two available thiols polymerize through the formation of disulfide bridges, in very mild and selective conditions, without protecting the catechol moieties, using iodine [50]. Following this approach, colorless and hydrophobic coatings have been obtained in flat substrates of a different nature, as well as on different textile pieces that were successfully used as filters to capture oil spills in water. Moreover, the alkyl can be replaced by a fluorescent tag. In addition to demonstrating the multifunctionality of our approach, the controlled combination of different fluorescent tag/alkyl-chain ratios along the polymerization process allows for the obtaining of coatings with control over the WCA. We will also demonstrate that the presence of catechol units is fundamental for good adhesion, which we will do by synthesizing and testing blank coating where the catechol moiety has been replaced by a phenylethenyl fragment.

## 2. Materials and Methods

### 2.1. General Procedures

The high quality solvents, chemicals, and reagents were acquired without any need for further purification from various commercial chemical companies such as Merck (Darmstadt, Germany), Scharlab (Sentmenat, Spain), Apollo Scientific (Cheshire, UK), Alfa Aesar (Kandel, Germany), and TCI (Zwijndrecht, Belgium). All reactions were monitored using analytical thin-layer chromatography (TLC) using silica-gel-60-precoated aluminum plates (0.20 mm thickness). Flash column chromatography was performed, using silica gel Geduran^®^ SI 60 (40–63 µm). The ^1^H NMR and ^13^C NMR spectra were recorded at 298 K at 250, 360, 400 MHz and 90, 100 MHz, respectively. Proton chemical shifts are reported in ppm (δ) (CDCl_3_, δ 7.26 or CD_3_COCD_3_, δ 2.06 or CD_2_Cl_2_ δ 5.32). Carbon chemical shifts are reported in ppm (δ) (CDCl_3_, δ 77.16 or CD_3_COCD_3_, δ 29.8 and CD_2_Cl_2_ δ 53.5. Infrared spectra (IR) were recorded on a Bruker Tensor 27 Spectrophotometer equipped with a Golden Gate Single Refraction Diamond ATR (Attenuated Total Reflectance) accessory. Peaks are reported in cm^−1^. HRMS were recorded in an Agilent 6454 Q-TP spectrometer with an Argilent Jetstream Technology (AJT) source, using electrospray ionization (ESI) or electronic impact (EI).

### 2.2. Synthesis of Monomers

The synthesis of all monomers and their full chemical characterization is fully described in the supporting information, following a synthetic procedure already described in the literature [51,52].

### 2.3. Polymerization Reactions

#### 2.3.1. Ex Situ Polymerization in Solution

As a general procedure, 1.1 equiv. of a solution of 35 mM of resublimed iodine in EtOH 96% was added dropwise to a ~7 mM solution of functional bis-thiol building blocks (**4a**, **4b**, **6a**, and **6b**) in EtOH 96%. The reaction mixture was stirred for 1 h at rt, after which a yellowish solid precipitated. The supernatant was decanted, and the solid washed with fresh EtOH 96% three times, and dried under vacuum.

In all cases, the monomeric bis-thiol building blocks contained 3% of the related catecholic tris-thiol (**2** for the building blocks **4a** and **4b**, **7** for the building blocks **6a** and **6b**). Polymers (yield): **P4a** (34%), **P4b** (39%), **P6a** (29%), and **P6b** (60%). Copolymers bearing fluorescent and hydrophobic building blocks **4a** and **4b**, respectively. Both monomers **4a** and **4b** were dissolved in EtOH 96%, and a 35 mM solution of iodine (1 equiv.) in EtOH 96% was added dropwise. After 1 h of stirring at rt, the precipitated derivatives were isolated, washed several times with fresh EtOH 96%, and dried with a gentle flux of N_2_. Yields were calculated based on the final amount obtained of the resulting copolymer vs. the initial ones used from both starting monomers. Copolymers: **C(4a-4b)a** [20% **4a**, 80% **4b**; 64% yield]; **C(4a-4b)b** [40% **4a**, 60% **4b**; 64% yield]; **C(4a-4b)c** [40% **4a**, 40% **4b**; 64% yield]; and **C(4a-4b)d** [30% **4a**, 60% **4b**; 65% yield].

#### 2.3.2. In Situ Polymerization

A total of 50 mg of resublimed iodine was placed at the bottom of a 20 mL vial. Test slides coated with the C_18_-functionalised monomer was placed face-down on top of the vials, and left standing overnight. The slides were then washed three times in EtOH 96% (3 × 3 mL) to remove the excess of adsorbed iodine, and dried in a gentle flux of Ar.

### 2.4. Polymers Characterisation

The products obtained from polymerization reactions were characterized by different techniques, such as ^1^H NMR, DOSY NMR experiments and gel permeation chromatography (GPC). The ^1^H NMR and DOSY experiments were performed in THF-d_8_, whereas the GPC analyses were performed in THF. A first rough value of the polymerization degree in the material was determined by ^1^H NMR, calculating the ratio of the thiol peak of the starting monomer and the resulting product from the polymerization reaction. For some of the materials, their polymerization degrees were also determined by calculating the corresponding molecular weights by means of GPC and DOSY NMR techniques. To prepare the GPC samples, the derivatives obtained after polymerization were dissolved in THF (1 or 2 mg/mL) and filtered through 0.22 µm nylon filters. To determine the polymerization degree by DOSY NMR experiments, firstly a calibration curve was needed. Hence, a pattern of 1-octadecene was used. Its diffusion coefficients were measured, and the diffusion coefficients of the products obtained from the polymerization reactions were interpolated into the calibrate curve to obtain the approximate values of their molecular weights.

The molecular weight distribution of all polymers was determined by gel permeation chromatography (GPC) using an Agilent Technologies 1260 Infinity chromatograph and THF as a solvent. The instrument is equipped with three gel columns: PLgel 5 μm Guard/50 × 7.5 mm^2^, PLgel 5 μm 10,000 Å MW 4 K−400 K, and PL Mixed gel C 5 μm MW 200−3 M. Calibration was carried out by using polystyrene standards. In each experiment, the freshly prepared polymer-sample of interest was dissolved in THF (1–2 mg/mL), and immediately analyzed using GPC (1 mL/min flow; 30 °C column temperature). The values obtained for Mn are an approximation.

The contact angle of the Milli-Q water droplets (ca. 5 μL) on the coated substrates was used to evaluate the hydrophobicity of the coated samples at rt, using the sessile-drop technique. An Easy Drop Standard analyzer and the Drop Shape Analysis DSA 10 software (KR Ü SS GmbH, Hamburg, Germany) were used throughout. The reported values arise from averaging CA measurements on three different spots of each sample. Energy-dispersive X-ray (EDX) line-scan profiles were obtained at rt and 200 kV on a FEI Tecnai G2 F20 coupled to an EDAX detector. Surface-topography imaging of the different samples was carried out in ambient air, in tapping mode using beam-shaped silicon cantilevers (Nanosensors, nominal force constant: 5 N·m^−1^, tip radius: ~7 nm) on an Agilent 5500 AFM/SPM microscope (Keysight Technologies, Santa Clara, CA, USA) combined with PicoScan5 version 1.20 (Keysight Technologies) software. An external X-Y positioning system (closed loop, NPXY100E from nPoint, USA) was used. Image processing was carried out using open-source software: WSxM version 3.1 (Nanotec Electronica, Madrid, Spain) and Gwyddion version 2.46 (CMI, Brno, Czech Republic). Scanning-electron-microscopy (SEM) measurements were carried out on a Quanta FEI 200 FEG-ESEM microscope operating at 20 kV. All samples were fixed on SEM holders. Prior to observation with SEM, all samples were metallized with a thin 15 nm layer of platinum using a sputter coater (Leica). Optical and fluorescence images were recorded on a Zeiss Axio Observer Z-1 inverted optical/fluorescence microscope, equipped with five different magnification lenses (5×, 10×, 20×, 50× and 100×), a motorized XY stage, Hg-lamp excitation source (HBO 103/2, 100 W), AxioCam HRc digital camera, and standard filters in fluorescence mode with an Alexa Fluor 488 filter.

### 2.5. Coating Experiments

Coating of solid macroscopic substrates. All substrates were cut into 1.5 × 1.5 cm square slides. Aluminum, copper, and stainless-steel slides were cleaned by sonicating successively in HPLC-grade acetone, EtOH 96% and Milli-Q water (10 min each), and dried in a gentle flux of Ar. Glass slides were cleaned with a plasma cleaner machine (400 W, 5 min). Clean slides were submerged in a ~7 mM solution of the corresponding derivatives in HPLC-grade CH_2_Cl_2_ (for **P4b** and **P6b**), or HPLC-grade acetone (4 mL) (for **P4a** and **P6a**), and left overnight without stirring. Finally, the slides were washed three times with fresh HPLC-grade solvent (CH_2_Cl_2_ or acetone), and dried in a gentle flux of argon.

Coating of cotton and polyester cloths. A piece of ca. 1.5 × 1.5 cm of cotton and polyester cloths (σ = 25 mg/cm) were submerged in a ~7 mM solution of the corresponding derivative in HPLC-grade acetone (for **P4a**), or HPLC-grade CH_2_Cl_2_ (**P4b** and **P6b**) (4 mL), and left overnight without stirring. The coated textiles were then washed with fresh HPLC solvent (CH_2_Cl_2_ or acetone) (3 × 2 mL), and dried in a gentle flux of Ar.

Static contact-angle (WCA) measurements on coated substrates (metals, glass, cotton, and polyester) were carried out at rt with Milli-Q water droplets (ca. 5 µL), by means of the sessile-drop technique. Reported values arise from averaging the CA measurements on three different spots of each sample.

Oil absorption in an oil-in-water (*o*/*w*) mixture. TDC, or olive oil (4 g) colored with Disperse Red 13 (200 ppm) was mixed with distilled water (15 mL). A piece of 1.5 × 1.5 cm cotton cloth (σ = 25 mg/cm, either pristine or coated with the C_18_-functionalised derivative **P4b**) was submerged in the *o*/*w* mixture for 15 s, taken out, and left hanging until the excess of liquid drained completely.

Filtration of an *o*/*w* mixture. A round piece (Ø ~ 2 cm) of cotton cloth (σ = 25 mg/cm), either pristine or coated with the C_18_-functionalised derivative **P4b**, was carefully placed on the mouth of a glass vial and folded slightly inwards. A 1:1 (*v*/*v*) hand-shaken mixture of distilled water and a solution of Disperse red 13 in Miglyol^®^ 840 was carefully added dropwise on the textile, until liquid started to filter through it.

Emulsion-breaking test. A solution of Disperse Red 13 (200 ppm) in Miglyol^®^ 840 (5 g) was mixed with a solution of sodium dodecyl sulphate (SDS) (150 mg) in distilled water (15 mL), and stirred with a Turrax^®^ homogeniser at 5000 rpm for 10 min at rt, yielding a homogeneous, milky, pink stock-emulsion. A piece of 1 × 1 cm of cotton cloth (σ = 25 mg/cm^2^, either pristine or coated with alkylated P4b), was submerged in a 10× diluted aliquot of the stock emulsion, gently stirred by hand for 5 min, and subsequently taken out of the treated emulsion and left to dry in air.

## 3. Results and Discussion

### 3.1. Synthesis and Characterization of Monomers

The modular synthetic strategy used to obtain the constitutive monomeric building blocks is depicted in the supporting information. In brief, starting from the pivotal tetrathiol pentaerythritol tetrakis(3-mercaptopropionate), **1**, the functional fragments of the monomers were sequentially introduced to obtain the catecholic functional monomers **4a** and **4b**, along with model monomers **6a** and **6b** (more information about the synthetic procedure in Appendix A).

### 3.2. Synthesis and Characterization of Polymers

Afterwards, the polymerization of monomers **4a** or **4b,** through the formation of disulfide bonds between the remaining thiol groups using mild and simple oxidative reaction conditions, was carried out (Figure 1). For this, a 7 mM solution of a given monomer in EtOH 96% was reacted with two equivalents of iodine (35 mM in EtOH 96%) for 3 h. at rt. It is worth mentioning that, to favor the crosslinking of the monomers, the reaction was also doped with a 5% of tristhiol **2**, affording the corresponding uncolored precipitates **P4a** or **P4b,** in moderate yield.

Following a similar procedure, the blank oligomers **P6a** and **P6b** were also synthetized as blank models. For details of the characterization and experimental conditions used in the all the synthetic steps, see Supporting Information. To determine the degree of polymerization, two techniques were used. Gel-permeation-chromatography (GPC) experiments in THF allowed for the determination of the apparent Mn and Mw, while diffusion-ordered-spectroscopy (DOSY) experiments were used to determine the diffusion coefficients of the polymerization products in THF-d_8_; from there, the corresponding M_n_ or M_w_ was determined, using a calibration-curve interpolation. Both techniques converged on similar and consistent values, indicating that the resulting polymerization products comprise oligomers of around 2–7 monomeric units (for more information about the synthetic procedure see Appendix A for the characterization).

### 3.3. Hydrophobic Coatings Experiments

#### 3.3.1. Hydrophobic Coating and Characterization of Different Material Surfaces

Once the polymeric materials were prepared and characterized, the first coating experiments were carried out by immersing 1.5 × 1.5 cm glass slides, previously cleaned with plasma (400 W, 5 min), in a ~7 mM CH_2_Cl_2_ solution of **P4b** overnight, without stirring. After this period, the substrates were removed from the solution and dried in a gentle flux of argon. The wettability of such surfaces was evaluated from WCA measurements, using Milli-Q water droplets (ca. 5 µL) by means of the sessile-drop technique. These measurements showed a contact-angle value of 95° (5° for the pristine glass). It is worth mentioning that the **P4b** robustness was challenged with intensive cleaning three times, with fresh CH_2_Cl_2_, in which the oligomer is soluble, in the presence of ultrasounds. After such a strong cleaning process, the substrate extraordinarily still retained a hydrophobic character, with a contact-angle close to 80°. For this reason, and to reinforce the viability of our approach, from now on all the substrates of this work were cleaned under these drastic conditions. For comparison purposes, the same procedure was now repeated using the precursor monomer **4b**, previous polymerization. In this case, even though the initial WCA value was relatively high (84°), it immediately dropped down to 55°, confirming that the monomer self-assembly coatings are less stable (for more information about the coating-procedure see Appendix A).

AFM surface-topography imaging of a standard **P4b** glass coating in ambient (air) conditions revealed a consistent average thickness of 1.5 µm and a roughness of 435 ± 32 nm, as measured upon scratching a section of the substrate (see Figure 1b). Other experimental factors such as substrate orientation and time evolution were also studied. For this, four cleaned glass-slides were submerged in a **P4b** HPLC-grade CH_2_Cl_2_ solution for 1 h, 4 h, 8 h, and overnight. Afterwards, they were washed three times with fresh CH_2_Cl_2_, dried in a gentle flux of argon, and the WCA was measured. It is worth mentioning that the WCA values after 4 h were comparable with the ones obtained after 8 h, or even overnight (the WCA values ranged between 71° and 76° in all cases). Therefore, and since it is the shortest experimental time used, experiments at 4 h were now repeated, depositing the slides into the solution with two different orientations, one lying on the bottom of the vial (parallel) and the second one resting on the wall (perpendicular). Afterwards, they were washed with fresh CH_2_Cl_2_, and dried with a flux of argon, resulting in comparable hydrophobic values (73° vs. 76°, respectively).

Beyond the glass, three additional surfaces (aluminum, copper, and stainless steel), were also studied, for comparison purposes. As a general procedure, 1.5 × 1.5 cm slides were cleaned by sonicating in acetone, EtOH 96% and Milli-Q water for 10 min, and dried in a gentle flux of argon. Afterwards, coating with **P4b** was achieved, following the same formula used for the glass surfaces, and characterized by EDX, showing in all cases signals around 2.3 KeV, characteristic of sulphur atoms (see Appendix A). Furthermore, carbon and oxygen were also identified, and the measured percentages of these were around the expected ones (±10%).

Afterwards, the WCAs were measured, and from there, two main deductions can be extracted (Figure 1). First, in all the cases our coating clearly improves the hydrophobic character, even if the pristine material already exhibits considerable WCA values (see Figure 1a). Second, the WCA differences before and after washing are even less remarkable than for glass, endorsing the robustness of our coating. In fact, once more, coating of the three substrates with the styrene derivative **P6b** revealed WCA values remarkably reduced, in comparison with **P4b,** except for copper substrates, most likely due to the affinity of unreacted terminal thiol-groups for this material. Finally, and regardless of the surface, no color modification was detected upon coating.

The polymerization protocol was also tested in situ. For this, the four substrates (including glass) were initially coated by submerging them in a ~7 mM CH_2_Cl_2_ solution of **4b,** and afterwards placing them on closed vials overnight with iodine, ensuring sublimation. Afterwards, surfaces were washed three times with fresh EtOH 96% to remove the excess of iodine, and dried in a gentle flux of argon. At first sight, the area in contact with iodine had already become scattered, pointing to a polymerization reaction. Accordingly, in all cases there was a considerable increase of WCA values, compared with the corresponding blanks (iodine sublimation in the absence of the **P4b** coating), and even for the corresponding coating obtained in solution. This did not apply for aluminum and especially stainless-steel, a fact attributed to the iodine effects (the direct interaction of stainless steel with iodine reduces its WCA from 44° to 12°). An EDX analysis of all samples showed the presence of sulphur at 2.3 keV, carbon at 0.27 keV, and oxygen at 0.52 keV.

#### 3.3.2. Hydrophobic Coating and Characterization of Textile Pieces

After testing the flat surfaces, the next objective was to confer hydrophobic features onto the textile pieces. As a general procedure, pieces of ca. 1.5 × 1.5 cm of cotton or polyester cloths, without previous treatment, were submerged in a ~7 mM **P4b** CH_2_Cl_2_ solution for four hours, without stirring. The coated textiles were then washed with fresh CH_2_Cl_2_ (3 × 2 mL), and dried in a gentle flux of argon. Analysis of the cotton and polyester fibers before and after coating using scanning-electron-microscopy (SEM) images, revealed remarkable differences between them, attributed to the formation of a coating around each one of the individual fibers. It is worth mentioning that no noticeable color nor flexibility and breathability changes of the textiles were macroscopically detected. On top of this, the WCA of Milli-Q water droplets at rt by means of the sessile-drop technique revealed values of 120° (137° before strong washing) for cotton@**P4b,** and 127° (163° before strong washing) for polyester@**P4b** (WCA values for pristine cotton and polyester were below 10° after a few seconds). Moreover, water droplets stand for a long period without remarkable changes. On the contrary, styrene derivative **P6b**-coatings of both cotton fibers and polyester were completely removed with washes, showing final WCAs close to 0° (for more information see Appendix A).

#### 3.3.3. Oil-Absorbance and Oil/Water-Separation Experiments with Hydrophobic Coated Cotton-Textile

Because of its remarkable water-repellent performance, **P4b**-coated textiles were then used for oil-absorbance tests and oil/water-separation experiments, simulating the removal of oily pollutants from aqueous phases. Two oily phases, tetradecane (TDC) and olive oil, colored with Disperse red 13, were used as non-volatile model pollutants upon the addition of 15 mL of distilled water. Afterwards, **P4b**-coated cotton fibers of known weight (dry) were soaked in the oil phase for 15 s, taken out, allowed to drain for 3 h, and weighted again. The coated fibers showed a 127% weight increase with TDC (25% for pristine cotton), and a 172% weight increase with olive oil (94% for pristine cotton). In accordance with this, SEM images of the fibers before and after absorbing TDC and olive oil reveal differences. A piece (1 × 2 cm) of cotton@**P4b** was also placed on top of a 20 mL vial, and used as a filter for phase separation of a Miglyol^®^ 840 colored with a red-Disperse-13 and distilled-water mixture (1:1). Immediately after deposition with a syringe, oil quickly saturated the coated-cotton passing through, while water was retained at the top. Another piece of cotton@**P4b** (1 × 1 cm) was also successfully used to recover a microemulsion (20 µm average droplet-size) of Miglyol^®^ 840 colored with Disperse red 13 in water stabilized with sodium dodecyl sulfate (SDS). For this, the textile was submerged in a 10× diluted aliquot of the stock emulsion, gently stirred by hand for 5 min, and subsequently taken out of the treated emulsion and left to dry in air. The coated cotton absorbed up to 97% of its own weight, and therefore acquired a remarkable red color, whereas the weight increase for the uncoated cotton was only 8% (Figure 2).

#### 3.3.4. Fluorescent Coating and Characterization of Glass Surface and Textile Pieces

We also tested our approach by replacing the alkyl chains with the fluorescein derivatives **4a**, **P4a** and **P6a,** using the experimental procedure previously described, but replacing the CH_2_Cl_2_ with acetone. After rinsing the slides well, the fluorescent images of an inverted microscope using an Alexa Fluor 488 filter revealed that only the **P4a**-coated glass showed continuous intense fluorescence on the whole surface. The **4a**-coated glass slide showed low fluorescent intensity concentrated in some aggregates, while the styrenic derivative **P6a** was completely removed, as the final surfaces did not show fluorescence. Similar results were obtained by repeating the protocol with cotton and textiles, as shown in Figure 3.

#### 3.3.5. Dual-Modulated Hydrophobic and Fluorescent Coating of Textile Pieces

In addition to a multifunctional character, the fluorescent groups also provided higher polarity (WCAs of 31° and 34° for cotton and textiles, respectively), a fact that was then used to modulate the hydrophobic/hydrophilic balance of the textiles. For this, oligomers with different and defined **4a**/**4b** monomer ratios (80:20, 60:40, 40:60 and 20:80) were obtained in good yields (close to 65% in all the cases) following the same protocol previously described for **P4a** and **P4b** (see Supporting Information for experimental details and characterization). Afterwards, cotton fibers were coated with the resulting **C(4a-4b)** oligomers from different ratios upon immersion in ~7 mM solutions in CH_2_Cl_2_ HPLC-grade for 4 h. Once removed from the solution, they were washed three times with fresh acetone/CH_2_Cl_2,_ and dried in a gentle flux of argon. Interestingly, WCAs systematically increased from 0° to 120°, with the amount of the aliphatic derivative **4b** present in the **C(4a-4b)** oligomer (see Figure 3).

## 4. Conclusions

We designed a novel catechol-based modular synthetic approach to control the wettability of surfaces in a straightforward and systematic manner, using colorless coatings. For this, we used a unique basic scaffold, pentaerythritol tetrakis(3-mercaptopropionate) **1**, conjugated with both a catechol unit and a functional group of two thiol groups, while leaving the two additional free thiols available for polymerization, through the formation of disulfide bridges. As a proof of concept, we synthesized oligomers that confer a hydrophobic and/or fluorescent character to the surface of glass slides and cotton/textile weaves. Hydrophobic fabrics were, in fact, successfully tested on simulated oil-spill and emulsion samples. Moreover, the proper selection and combination of building block units combining both functionalities allowed us to systematically fine-tune at will the wettability of surfaces. All in all, the modular character of our synthetic approach and its rich and flexible chemistry open new opportunities for the development of colorless coatings with tailored properties. This is so thanks to the presence of the catechol moiety, which plays an important role in the adhesion processes, resulting in robust coatings even after vigorous washing processes.

## Data Availability

Not applicable.

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
