# Peer review of "Mussel-Inspired Lego Approach for Controlling the Wettability of Surfaces with Colorless Coatings"

_biomimetics, 2022, doi:10.3390/biomimetics8010003_

Round 1

Reviewer 1 Report

The paper deals with importnat topic which is a prepration of new multifunctional coatings based on catechols. The author showed very smart synthetic routes towards a new catchol based polymers with sulphur moieties. The experiments demonstarting the adhersive character of the polymesr  as well as their characterization is well presneted and the conslusion are supported by the obtined resluts. Please just clarify the size of of materials pieces. I can imaging  that they were rectangles 1.5x1,5 cm (lengh X width) but what do you mean by 1.5x,1.5 cm2. Did they have a surface area of 1.5 cm2??I also recomned to imprved a grphics/graph since they do not look profesional. I assumed they were made in Excel. Try to uniform the appearance of the graphics and used origin or graphpad to make them more attractive.

Author Response

The paper deals with importnat topic which is a prepration of new multifunctional coatings based on catechols. The author showed very smart synthetic routes towards a new catchol based polymers with sulphur moieties. The experiments demonstarting the adhersive character of the polymesr  as well as their characterization is well presneted and the conslusion are supported by the obtined resluts.

We do really appreciate the comment. Thank you

Please just clarify the size of of materials pieces. I can imaging  that they were rectangles 1.5x1,5 cm (lengh X width) but what do you mean by 1.5x,1.5 cm2. Did they have a surface area of 1.5 cm2??

The referee is right, thanks for the comment. All the area units have been properly revised and corrected

I also recomned to imprved a grphics/graph since they do not look profesional. I assumed they were made in Excel. Try to uniform the appearance of the graphics and used origin or graphpad to make them more attractive..

Thanks for the comment and although we may understand it, we do prefer to leave graphs as they are cause even though they may not seem appealing, turn out to be quite clear for the audience.

Reviewer 2 Report

The manuscript under consideration deals with a biomimetic approach to control the wettability of different coated surfaces. The paper is well organized in proper sections. Abstract and conclusions well define aims and findings of the work. The experimental section, especially regarding wettability, should be improved including contact angle hysteresis which in such conditions, could help to interpret the results

Author Response

The manuscript under consideration deals with a biomimetic approach to control the wettability of different coated surfaces. The paper is well organized in proper sections. Abstract and conclusions well define aims and findings of the work.

We thank very much this reviewer for the kind appreciation of our work

The experimental section, especially regarding wettability, should be improved including contact angle hysteresis which in such conditions, could help to interpret the results

We thank referee observation. Unfortunately, we do not have a proper setup to measure advancing and receding contact angles. In any case, this experiment brings information about the chemical and topographical heterogeneity of the surface, information that is already given even more accurately in the manuscript with AFM topography and SEM studies.